# Skin cancer classification using novel fairness based federated learning algorithm

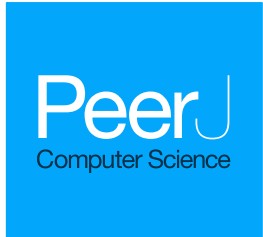

Awais Karni[1], Qamar Abbas[2], Jamil Ahmad[3] and Abdul Khader Jilani Saudagar[4]

[1] Computer Science, International Islamic University, Islamabad, Pakistan
[2] Computer Science, International Islamic University, Islamabad, Pakistan, Pakistan
[3] Member IT, HEC, Islamabad, Pakistan
[4] Information Systems Department, College of Computer and Information Sciences, Imam Mohammad Ibn Saud Islamic University (IMSIU), Riyadh, Saudi Arabia

Corresponding author
Abdul Khader Jilani Saudagar,
aksaudagar@imamu.edu.sa

## ABSTRACT

Numerous skin conditions fall under the category of dermatological diseases, which make proper diagnosis and treatment planning difficult. Our research centres on tackling these obstacles within the framework of federated learning, a decentralized approach to machine learning. We provide a unique strategy that combines class-weighting strategies to reduce the negative effects of different data distributions among decentralized clients by leveraging the federated average algorithm. We assessed the effectiveness of our approach using the Fitzpatrick 17k dataset, an extensive collection encompasses a wide range of skin conditions. With its realistic representation of dermatological diagnosis scenarios, the dataset provides a solid foundation for training and testing federated learning models. One of the main issues driving our research is the ubiquitous problem of class imbalance within federated learning. When client data distributions are uneven, class imbalance can result in biased model predictions and subpar performance. To solve this issue and enhance model performance, we have incorporated class-weighting approaches into the federated average architecture. We show through thorough experimentation that our strategy is useful for improving federated learning models' learning performance. Our methodology presents a possible solution to the class imbalance issue in federated learning situations by reducing bias and increasing prediction accuracy. Our study further emphasizes the significance of iterative refinement methods for optimizing federated average weights and fine-tuning model parameters. The results of our study show that the model performance has improved significantly, with an average accuracy of almost 92% across all categories. These results highlight our classification model's potential usefulness for dermatological diagnosis and treatment planning in clinical settings. Furthermore, this study contributes valuable insights into the application of federated learning for dermatological disease classification, paving the way for future advancements in addressing key challenges such as data privacy, distribution heterogeneity, and model fairness in medical imaging.

# INTRODUCTION

In machine learning, "learning" refers to the model's ability to enhance its predictions by minimizing a loss function that quantifies the error between its predictions and actual values (*Sekeroglu, Dimililer & Tuncal, 2019*). This process is divided into supervised, unsupervised, and semi-supervised learning. Supervised learning involves training with labeled data to map inputs to outputs for tasks like classification and regression (*Ogbuabor & Ugwoke, 2018*). Unsupervised learning, using only unlabeled data, discovers hidden patterns and structures, such as through clustering and dimensionality reduction. Semi-supervised learning merges a small amounts of labeled data with a large amounts of unlabeled data to refine its predictive capabilities. Centralized learning trains models on aggregated data on a central server, simplifying training and managing sensitive information more effectively (*Savić et al., 2021*).

Centralized learning relies on a central server and stable internet, which can be problematic in areas with poor connectivity and poses security risks due to centralized data storage. Strong encryption and authentication can mitigate these risks, making centralized learning effective for large datasets. In contrast, decentralized (federated) learning allows multiple parties to train models without sharing raw data, enhancing privacy and collaboration (*Korkmaz et al., 2020*). Each party updates its local model and shares parameters with a central server, which aggregates them to improve the global model. This method benefits privacy-sensitive fields but faces challenges like high communication costs and potential vulnerabilities to attacks on the central server.

Federated learning (FL) trains models across multiple devices without centralizing data, enhancing privacy (*Roth et al., 2020*). Local models are updated on-device and then combined into a global model, iteratively improving accuracy. This technique is useful for sensitive data in fields like healthcare and finance, allowing secure, decentralized analysis and model training. Imbalanced data occurs when one class is significantly underrepresented, such as in fraud detection, where valid transactions outnumber fraudulent ones (*Wu, He & Chen, 2020*). This can cause models to favor the majority class and poorly predict the minority class. Solutions include oversampling the minority class, undersampling the majority, and using specialized algorithms or metrics for imbalanced data.

Cancer is a very dangerous disease for human life, and skin cancer is considered one of the common cancer in the world (*Roky et al., 2025*). The early detection of skin cancer is considered very helpful in curing this disease. The machine learning models are widely by the researchers for the analysis of skin images for various skin diseases, including cancer. Federated learning models are widely used by researchers to solve various real-world problems, including medical imaging problem solving using data of multiple clients (*Nasajpour et al., 2025*). Federated learning performs well on the balanced datasets; however, these models face certain challenges for the clients' imbalanced and not independently and identically distributed datasets. By considering the scenario of skin cancer, one of the hospitals in a specific region can handle a higher number of cases due to high ultraviolet exposure, and any other hospital may observe a small number of cases due

to less ultraviolet exposure. Due to a lack of fairness awareness, the federated learning models dealing with such types of situations face performance degradation and unstable convergence issues, leading to limitations of this problem-solving approach (*Fan et al., 2025*). This was the main motivation considered in this research work to explore a fairness-based approach focusing on the data sample contribution by various clients.

Irregular or uneven data distribution, known as 'imbalanced data,' occurs when one class or category is significantly more prevalent than others within a dataset (*Carvalho, Pinho & Brás, 2025*). A common example is in fraud detection, where most transactions are legitimate, with only a small fraction being fraudulent (*Duan et al., 2020*). This imbalance poses challenges for standard machine learning algorithms, which typically prioritize overall accuracy by favoring the majority class, potentially ignoring the minority class. This bias can lead to models that are insensitive to important minority class instances. Additionally, imbalanced data can exacerbate overfitting issues, where a model becomes overly tailored to the training data and fails to generalize effectively to new data. Various strategies exist to address imbalanced data, such as oversampling the minority class to increase its representation or undersampling the majority class to balance the dataset (*Chakraborty & Ghosh, 2022*). Algorithms for imbalanced data, such as decision trees, and adapted evaluation metrics offer viable solutions. The choice of approach should be guided by the specific characteristics of the dataset and the objectives of the application.

## Problem statement

Federated learning is considered a very effective privacy-preserving and knowledge sharing problem solving approach; however, its performance is affected by the imbalanced and non-Independent and Identically Distributed (non-IID) data distribution across contributing clients, such as in real-world skin disease datasets. This distribution leads to bias in model performance, showing high performance for the majority classes and low performance for the minority classes in the datasets. This disparity is especially problematic in applications like skin disease detection, where underrepresented lesion types or skin tones may result in missed or incorrect diagnoses for vulnerable populations.

## Research objectives

The major goal of this research was to develop and evaluate methods for addressing class imbalance in datasets to improve the performance of machine learning models on imbalanced datasets:

- To develop and implement a fairness-aware federated learning algorithm that dynamically adjusts client contributions.
- To compare the performance of the proposed fairness-aware federated learning algorithm with other algorithms using a real-world skin image dataset such as Fitzpatrick17k.
- To assess the proposed fairness-aware federated learning model using a comprehensive set of performance metrics—including accuracy, area under the receiver operating

characteristic curve (AUC-ROC), precision, recall, and F1-score across multiple client distributions and skin disease categories.

- To test the significance of the proposed fairness-aware federated learning algorithm using a statistical significance test by considering all performance metrics.

## LITERATURE REVIEW

Federated learning is a decentralized paradigm for model training that makes it possible to collaborate and learn across dispersed nodes/locations. Federated learning allows collaborative model training while keeping data private on individual devices. FedAvg, a common method, aggregates models with fixed weights, but struggles with non-i.i.d. data (data distributions varying across clients). The study introduces Auto-FedAvg, which dynamically adjusts aggregation weights based on data distribution and training state. It improves model performance on datasets like CIFAR-10 and medical image analysis for COVID-19 and pancreas segmentation. However, Auto-FedAvg requires stable server-client connections, posing challenges in settings with frequent device changes, which could be mitigated by reducing communication frequency (*Xia et al., 2021*). Various aspects of the literature review are discussed in the following subsections.

### Federated learning in medical imaging

The research work (*Xu et al., 2022*) addresses global dermatological challenges, advocating for early diagnosis through smartphone apps using federated learning for privacy-preserving model aggregation. Their fairness-aware framework improves diagnostic accuracy across diverse skin tones, validated on the Fitzpatrick 17k dataset. The article introduces FL as a solution for privacy-preserving machine learning, emphasizing its potential for collective intelligence while protecting data privacy. It highlights challenges, especially involving business competitors in FL federations, which can lead to delays and unequal benefit distribution. To address these issues, the authors propose the FLI payoff-sharing scheme, designed to maximize collective benefit and mitigate inequalities among data owners, supported by experimental comparisons with existing methods (*Truex et al., 2019*). The authors present FedCM, a federated learning framework for early detection and classification of Alzheimer's disease (AD) using non-invasive data. FedCM employs model distillation to maintain model heterogeneity and prevent privacy breaches, focusing on sharing predictions rather than raw data or model weights. Tested on sMRI data from multiple datasets, FedCM outperforms previous FL and centralized learning systems in accuracy metrics and attention visualization on relevant brain regions, showcasing its viability in AD classification while addressing data privacy and distribution variability challenges (*Huang, Yang & Lee, 2021*). This study combines an upgraded version of the faster R-CNN with FL to propose a method for identifying multiple pests in orchards. FL allows data integration from various parties without centralizing data, reducing communication costs. Using ResNet-101 in the faster R-CNN improves detection accuracy for small targets, while multi-size fusion of feature maps enhances detection accuracy for pests and diseases of various sizes. The authors introduce the Soft-NMS

algorithm to address shading issues, achieving 90.27% average accuracy and reducing detection time to 0.05 seconds per image. FL further boosts mean average precision (mAP) to 89.34% while decreasing model training time by 59%. Future work includes refining the FL algorithm, adding adaptive strategies to improve accuracy and convergence rates, and enhancing dense object detection (*Deng et al., 2022*). The research work (*Wu et al., 2021*) presents a federated contrastive learning (FCL) framework for dermatological disease diagnosis on mobile devices, integrating FL with contrastive learning (CL). FCL enables pre-training on distributed, unlabeled data followed by fine-tuning on limited labeled data without compromising privacy. By sharing characteristics during pre-training across devices, the approach enhances recall and precision for diagnosing dermatological diseases across varied skin tones. Experimental results show significant improvements over existing techniques, underscoring the effectiveness of FCL in enhancing diagnostic accuracy and privacy preservation. FedHealth (*Chen et al., 2020*) is a federated transfer learning system designed for wearable medical devices, combining data from different organizations while preserving privacy. It uses federated learning to create personalized models, achieving a 5.3% accuracy increase in human activity recognition over baseline methods. This system enhances wearable healthcare, especially for diagnosing conditions like Parkinson's disease, by integrating large volumes of health data with advanced machine learning, addressing data fragmentation and model personalization issues. Federated transfer learning (FTL) (*Saha & Ahmad, 2021*) tackles data isolation and privacy in AI by allowing knowledge transfer across different user bases while maintaining privacy. FTL's applications and future directions emphasize the need for advanced techniques and real-world datasets to improve effectiveness. Secure FTL advances efficiency and security in data federation by using secret sharing (SS) and secure multiparty computing (MPC) with the SPDZ protocol. This model enhances collaborative training efficiency and protects against malicious actors, reducing runtime and communication costs significantly (*Sharma et al., 2019*).

## Federated learning for real-world problem-solving

The FOLB algorithm (*Nguyen et al., 2020*), designed for distributed mobile devices, achieves rapid convergence and improved accuracy by addressing the statistical heterogeneity of the system. Its adaptive aggregation effectively minimizes loss, reducing the number of communication rounds needed. Future research is suggested to refine device selection strategies to enhance performance. The research work by *Liu et al. (2020)* introduced a framework for privacy-preserving machine learning in decentralized data environments. It enables knowledge transfer among enterprises while protecting data privacy using techniques like homomorphic encryption and secret sharing. The framework facilitates accurate model generation without data exposure, outperforming traditional federated learning and achieving accuracy comparable to non-privacy-preserving methods. Future research aims to improve performance through distributed computing, enhancing the scalability of large data federations. FL is a promising solution for Internet of Things (IoT) networks, addressing distributed and privacy-sensitive data challenges (*Khan et al., 2021*). This approach, essential for applications in smart industries, intelligent

transportation, and healthcare, leverages advances in FL, focusing on robustness, privacy, and communication efficiency. Integrating FL with emerging 6G networks could enhance IoT applications while ensuring privacy and security. Current FL systems are analyzed, detailing their functional design, distributed training methods, and data processing techniques. A four-layer architecture covers presentation, user services, FL training, and infrastructure. The study discusses central, hierarchical, and decentralized aggregation methods and data manipulation strategies like as compression and RPCs. FL systems such as TensorFlow Federated and FATE are reviewed, with research areas, including interpretability and handling unbalanced data highlighted for future exploration (*Liu et al., 2022*). Federated learning faces challenges with non-IID datasets, particularly in domains like medical imaging and object detection. The article introduces model-contrastive learning (MOON) to enhance FL performance on non-IID datasets through contrastive learning at the model level. MOON improves collaboration in multiparty training without raw data exchange, showing superiority over current methods in diverse image categorization tasks. Its potential extends beyond vision-related problems, addressing the heterogeneity of local data distribution effectively (*Li, He & Song, 2021*).

## Fairness in federated learning

In the context of FL, the author explores fairness challenges beyond privacy and communication costs. They introduce AgnosticFair, a fairness-aware FL architecture that addresses unknown testing data distributions using kernel reweighing functions. This approach ensures fairness and high accuracy across diverse local datasets, demonstrating effectiveness on real-world data scenarios. Future research aims to expand this framework with additional fairness considerations and optimized kernel functions (*Du et al., 2021*). Auto-FedAvg addresses the dynamically adjusting weights based on data distribution and training progress. It demonstrates improved performance over existing FL methods in both general and medical image analysis tasks (*Xia et al., 2021*). A novel algorithm is introduced for fair resource allocation in federated learning, evaluating participants' contributions to model performance without sharing data. It uses weighted accuracy improvements to allocate resources proportionally, surpassing traditional methods in fairness and scalability for large-scale applications (*Li et al., 2019*). Federated learning coordinates model training across decentralized clients to preserve data privacy, tackling challenges like statistical heterogeneity and limited communication bandwidth. FedFa, a proposed algorithm combining double momentum gradient and weighting techniques, enhances convergence and fairness in heterogeneous networks (*Huang et al., 2020*).

## Machine learning deep learning models for medical image classification

*Bello, Ng & Leung (2024)* introduced a skin cancer diagnostic fine-tuned deep learning model in their research work. Their presented model was helpful to achieve better performance as compared to baseline models. *Yaman et al. (2022)* introduced a hybrid deep feature extraction model based on five pre-trained deep learning

models and an ImRMR-based feature selection model for skin cancer classification. *Baygin, Tuncer & Dogan (2022)* presented a discrete wavelet transform, local phase quantization (LPQ), local binary pattern (LBP), and pre-trained DarkNet models for feature extraction learning-based skin cancer classification model. The study (*Wang et al., 2021*) tackles long-tailed image classification, where data imbalance makes learning challenging. The authors propose a hybrid network combining cross-entropy loss for classifiers and supervised contrastive loss for feature learning. The method transitions from feature to classifier learning over time. They explore supervised contrastive loss (SC loss) and prototype supervised contrastive loss (PSC loss), with PSC being more memory efficient. Experiments on three datasets show the hybrid network's superior performance, making it the first study to apply supervised contrastive learning to long-tailed classification, with future work focusing on PSC loss optimization. *Tuncer et al. (2024)* introduced a fewer trainable parameters based lightweight CNN model in his research work. The authors (*Tao et al., 2019*) propose a novel self-adaptive support vector machine (SVM) cost-sensitive ensemble for imbalanced data classification, addressing challenges where the majority class overwhelms the minority samples. This approach enhances prediction accuracy by adjusting SVM decision boundaries to favor the minority classes, crucial for tasks like text categorization and intrusion detection. Their method, validated on real-world datasets, demonstrates robust performance and improves generalization by focusing on borderline the minority cases. Future directions include refining cost-sensitive strategies and optimizing boosting techniques to further enhance classifier performance in imbalanced datasets. The research work (*Tao et al., 2020*) proposes ACFSVM, an affinity and class probability-based fuzzy support vector machine, to handle imbalanced datasets. Traditional SVMs tend to favor majority classes, especially in noisy or outlier-rich datasets. ACFSVM uses an affinity measure computed in kernel space with an SVDD model on the majority class samples, coupled with class probabilities from kernel k-nearest neighbors. This approach identifies potential outliers and border samples in the majority class, emphasizing the significance of the minority class samples. Experimental results on UCI datasets show that ACFSVM outperforms existing methods in G-Mean, F-Measure and AUC, showcasing its effectiveness in imbalanced dataset classification. The study (*Lee, Jun & Lee, 2017*) addresses degradation issues in imbalanced data with their proposed method, which leverages SVM's ability to construct nonlinear boundaries and detect minor disjuncts and data shifts. They enhance SVM's performance in handling class imbalance by incorporating a weight adjustment factor into a weighted SVM used in AdaBoost. This factor focuses on borderline instances and positive noise, adjusting weights based on SVM margin categorization. Evaluation on 10 real-world datasets demonstrates the method's superior performance over standard SVM and various sampling and boosting techniques in terms of F-Measure and AUC. Ensemble learning techniques for classifying imbalanced data using metrics like accuracy, F1-score, g-mean, minutiae cylinder-code (MCC), Cohen's Kappa, and AUC, particularly in the context of cabbage image classification (*Wardhani et al., 2019*). Imbalanced data poses challenges in machine learning, especially in domains like medical and plant disease classification, where minority classes are underrepresented. Conventional metrics can be misleading, necessitating the use of

metrics like AUC, MCC, and Kappa to accurately assess classifier performance. Ensemble learning proves effective in mitigating imbalanced data issues, ensuring robust classification performance across various metrics.

## MATERIALS AND METHODS

### Materials

The experimentation and analysis, we have used an Intel Core i5 processor, 8 GB of RAM, and 500 GB computational resources for model training and evaluation. We have used the Windows-10 operating system, and the Google-Colab cloud service was used for the experimentation. All experiments, including model training and evaluation, were conducted on Google Colab using its cloud-based GPU environment for consistency and performance. The HP laptop was used solely for code development, preliminary testing, and result visualization. No model training or benchmarking was performed locally. For seamless execution, the Python environment in Colab was pre-configured with the necessary libraries such as TensorFlow, Keras, NumPy, Pandas, and Matplotlib. The data set used in this research work, is available at the link https://github.com/mattgroh/fitzpatrick17k (*Groh, 2021*). The Fitzpatrick17k dataset is used to evaluate the suggested approach. The Fitzpatrick17k dataset contains 16,577 images with six skin types, numbered 1 through 6, are labeled on the dataset, with smaller labels designating lighter skin and larger labels designating darker skin. Three separate categories make up the Fitzpatrick17k dataset: "Benign," "Malignant," and "Non-Neoplastic." Fitzpatrick17k sample images that illustrate benign, malignant, and non-neoplastic skin conditions are shown in Fig. 1.

### Methods

In federated learning, normally, data is divided into rounds. We have equally divided the training and testing datasets into 20 rounds. During the pre-processing, the dataset was divided into a training set (70%), a validation set (10%), and a test set (20%). The data used in the training, validation, and testing was also normalized. During the pre-processing, the dataset was loaded and then split the data based on Fitzpatrick scale values. To ensure consistency across images, the following pre-processing pipeline was applied. All images were resized to $224 \times 224$ pixels for compatibility with most CNN architectures. Pixel intensity values were normalized to the range [0, 1]. For the federated learning setup, a non-IID client partitioning strategy was applied to simulate realistic data heterogeneity across six virtual clients. The images available for the Fitzpatrick17k dataset were resized images and then we split these images to training, validation, and test sets as a part of pre-processing.

## PROPOSED FAIRNESS-BASED AUTOMATIC WEIGHT ADJUSTING FL FRAMEWORK

Our federated learning model integrates data from six geographically diverse hospitals to improve diagnosis accuracy while preserving privacy. By using DenseNet201, each hospital trains the model locally on their specific datasets over 20 rounds, and shares model weights

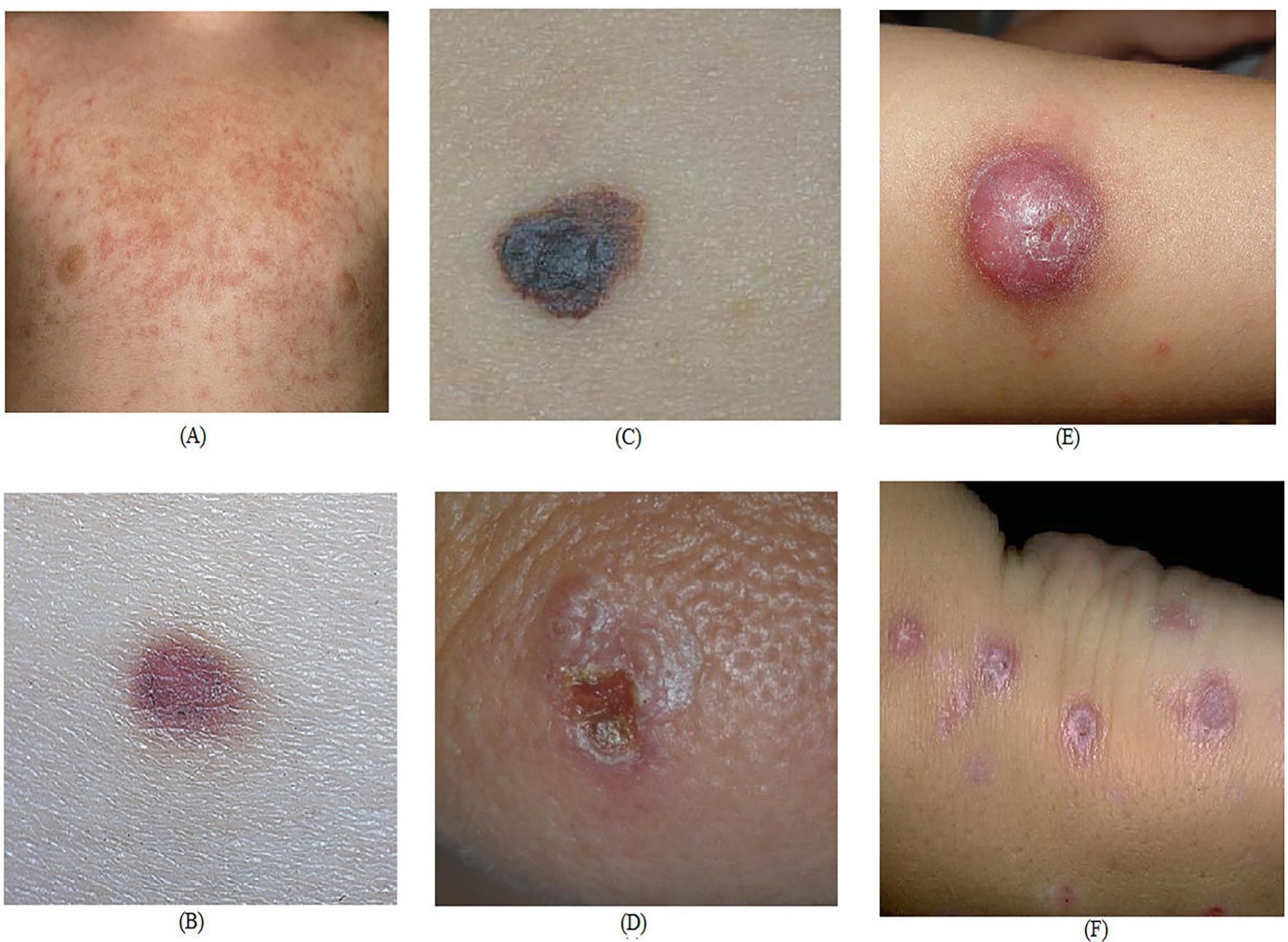

**Figure 1 Sample dermatological images Fitzpatrick17k dataset representation of benign (A, B), malignant (C, D), and non-neoplastic (E, F) skin lesions (*Groh, 2021*).**

for federated averaging, which enhances model robustness. The Fitzpatrick17k dataset is used to train the model *via* Google Colab and an HP laptop, employing image augmentation and transfer learning to refine the classification of dermatological conditions. This collaborative process improves model performance by addressing class imbalances and converging on a consensus model that leverages diverse data sources, with potential for broader applicability. The framework of fairness-based automatic weight adjusting federated learning is illustrated in Fig. 2.

## Automatic weight adjusting FL

The proposed algorithm is a federated learning approach designed to ensure fairness among participating clients while maintaining model performance. The list of parameters used in the algorithm is given in Table 1. The pseudocode of Algorithm 1 begins by initializing necessary parameters, including the client set $C$, the total number of

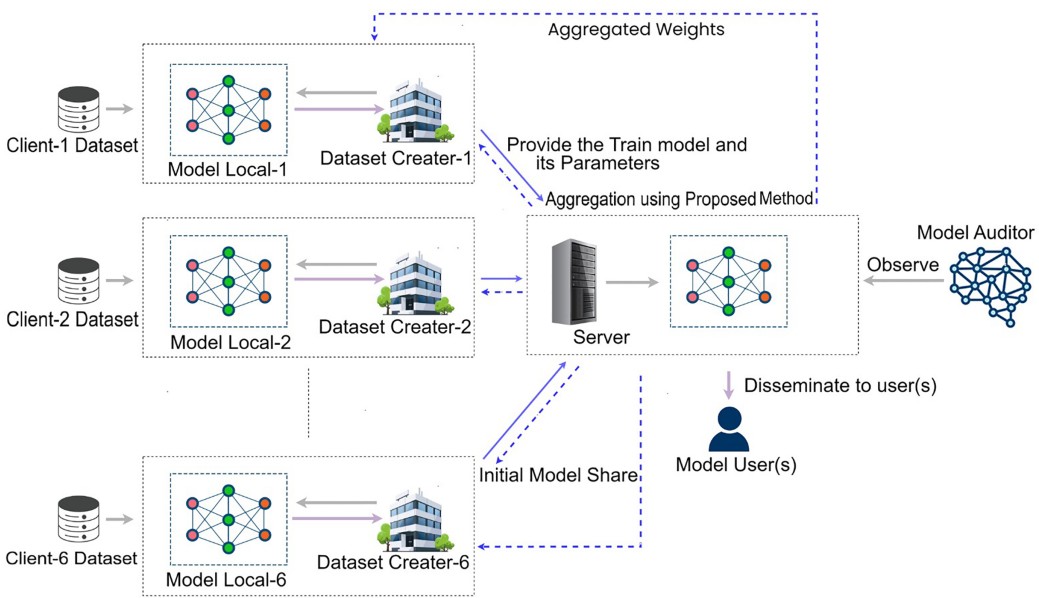

**Figure 2 Proposed fairness-based automatic weight adjusting FL(FbFedFAuto) framework.**

**Table 1 List of parameters used in the proposed algorithm.**

| Parameter | Description |
|---|---|
| $\lambda_c$ | Client relative contribution |
| m | Fairness adjustment counter |
| M | Upper bound of fairness adjustment |
| $\theta$ | The global model |
| B | Batch size |
| C | Client set |
| T | Communication rounds |
| E | Local epochs |

communication rounds $T$, the number of local epochs $E$, the batch size $B$, the global model $\theta$, a hyperparameter $Q$ that influences fairness weighting, and an upper bound $M$ for the fairness adjustment counter $m$, which is initialized to 1. For each communication round, the global model $\theta$ is sent to all clients. Each client initializes its local model $\theta_c$ with the global model and performs local training using the `LocalUpdate` function. During this process, the client splits its dataset into mini-batches of size $B$, iterates through $E$ local epochs, and updates the model parameters $\theta_k$ using gradient descent. The function also computes a fairness metric $\lambda_c$ based on the standard deviation of the class distribution within the client's data. A smaller standard deviation, indicating more balanced data, results in a higher fairness contribution. After local updates, the server collects the updated models and fairness metrics from all clients. If $m$, the fairness adjustment counter, is below the upper bound $M$, it is incremented by 1. The server then calculates fairness-based

---

**Algorithm 1** Proposed fairness-based contribution.

1: **Input:** $C$ is the client node, $T$ is each round, $E$ is the total number of epochs, $B$ is the batch size, $\theta$ is the local model, $Q$ is the hyperparameter, $M$ is the upper bound.

2: $m \leftarrow 1$

3: **for** each round $t \leftarrow 1$ to $T$ **do**

4:     **for** each client $c \in C$ **in parallel do**

5:         $\theta_c \leftarrow \theta$

6:         $\theta_c^\lambda \leftarrow \text{LocalUpdate}(c, \theta_c)$

7:     **end for**

8:     **if** $m \leq M$ **then**

9:         $m \leftarrow m + 1$

10:     **end if**

11:     **for** each client $c \in C$ **do**

12:         $w_c \leftarrow \dfrac{e^{m-\lambda_c}}{\sum_{i \in C} e^{m-\lambda_i}}$

13:     **end for**

14:     $\theta \leftarrow \sum_{c \in C} w_c \theta_c$

15: **end for**

16: **Function** $\text{LocalUpdate}(k, \theta_k)$

17: $B \leftarrow$ (split dataset of $k$ into batches of size $B$)

18: **for** each local epoch $i$ from 1 to $E$ **do**

19:     **for** batch $b \in B$ **do**

20:         $\theta_k \leftarrow \theta_k - \eta \nabla L(\theta_k; b)$

21:     **end for**

22: **end for**

23: $\lambda_c \leftarrow 1 - \text{stdev}(\text{class}_1, \ldots, \text{class}_n)$

24: **Return** $\theta_k, \lambda_c$ to server

---

weights $w_c$ for each client using an exponential weighting function, where the contribution of each client depends on $m - \lambda_c$. Clients with higher $\lambda_c$, signifying less fairness, contribute less to the global model update. Finally, the server aggregates the client models into the global model $\theta$ using the calculated weights. This algorithm is particularly suited for Federated Learning scenarios with heterogeneous and imbalanced data distributions. By incorporating a fairness metric into the aggregation process, it ensures equitable contributions from all clients, promoting balanced learning outcomes while preserving data privacy, as only models and fairness metrics are shared with the server. Class weighting is a key technique to address class imbalance in the model training, where minority class instances receive higher weights and majority class instances lower weights, balancing their influence. We refine this with a class imbalance ratio factor and an exponential parameter to fine-tune the weighting's impact, adjusted using the standard deviation of class frequencies. This ensures precise calibration of weights according to dataset characteristics, enhancing the model's ability to generalize and predict accurately across all classes by maintaining fairness and robustness in diverse applications.

## Experimental setup

In this research work, we have used the Fitzpatrick17k dataset for the skin cancer image classification problem using a federated learning technique. The goal is to cooperatively train a DenseNet201 architecture across several clients while maintaining data privacy by utilizing the federated average technique. Google Colab is used for the implementation, utilizing its powerful computational power to run the code quickly. An HP laptop is used to facilitate the experimentation process, providing a handy platform for overseeing and administering the federated learning process. The federated learning process consists of multiple essential steps in each iteration. To simulate a distributed learning environment, the dataset is first divided among several clients. To avoid exchanging sensitive data with third parties, the clients train their models independently on local data for every training cycle. This decentralized method fosters a cooperative model improvement while allaying privacy concerns. The models are trained using client-specific data for several epochs throughout the training phase of each round. By applying methods like image augmentation and transfer learning from DenseNet201 models that have already been trained, the models are taught to identify images into groups like benign, malignant, and non-neoplastic. Metrics like accuracy and loss are tracked throughout training to evaluate the model's convergence and performance. The federated averaging phase is essential for combining the knowledge from individual client models after local training. A consensus model that represents the combined intelligence of the participating clients is created by averaging the weights of the last layer across all clients. With the protection of privacy and secrecy, this federated averaging approach makes sure that the global model gains from the various data distributions found across various clients. The modified consensus model is then used as the basis for training on client data in the next rounds. The procedure is repeated several times, improving the model's performance through cooperative learning at each iteration. Extensive experimentation and analysis yield insights into the federated learning approach's scalability, performance improvements, and convergence behavior. All things considered, this study highlights the potential of federated learning in practical applications and its consequences for machine learning techniques that protect privacy. Client-wise partition of the specified dataset used in the experimentation is given in the Table 2. This non-IID partition simulates real-world data heterogeneity in federated learning environments for six federated clients.

## Transfer learning models, architectures and hyperparameters

Transfer learning is a machine learning technique where a pre-trained model, developed on one task, is adapted for use on another related task. This method is particularly beneficial when there is limited data available for the target task, as it leverages the knowledge learned from a larger dataset. In transfer learning, the model's architecture is typically retained while weights from the source domain are fine-tuned on a smaller target-domain dataset. This approach can significantly reduce training time and improve the model's performance, especially in tasks such as image classification, object detection, and natural language processing.

**Table 2 Client-level distribution of skin lesion categories across federated clients.**

| Client No. | Benign | Malignant | Non-neoplastic |
|---|---|---|---|
| Client 1 | 442 | 453 | 2,046 |
| Client 2 | 671 | 742 | 3,383 |
| Client 3 | 475 | 456 | 2,366 |
| Client 4 | 367 | 301 | 2,107 |
| Client 5 | 159 | 146 | 1,222 |
| Client 6 | 44 | 60 | 524 |

Transfer learning is commonly used with deep learning models, where the base model (*e.g.*, VGG16, ResNet50, DenseNet201) has already been trained on large datasets like ImageNet. The pre-trained model can be used either as a feature extractor or by fine-tuning some of its layers. This approach is now standard in machine learning, helping models generalize and handle limited labeled data.

### DenseNet201

Dense convolutional network (DenseNet) is a neural network architecture designed to strengthen feature propagation and encourage feature reuse by connecting each layer to every other layer in a feedforward manner. The DenseNet201 variant consists of 201 layers, including three dense blocks and transition layers that reduce dimensions through batch normalization, convolution, and pooling. Key hyperparameters include the growth rate, which determines the number of filters added per layer, and the compression factor in transition layers, which reduces feature maps. Optimizers like Adam or SGD are typically used with learning rates starting at 0.001 or 0.01. DenseNet201 is renowned for its compactness and ability to use fewer parameters without sacrificing performance.

### ResNet50

Residual Network (ResNet) addresses the vanishing gradient problem by introducing shortcut connections that bypass one or more layers, allowing gradients to flow through the network more effectively. ResNet50 comprises 50 layers divided into four residual block stages, each consisting of a convolution, batch normalization, and ReLU activation layers. The architecture starts with a small number of filters and increases their depth progressively. Common hyperparameters include the optimizer (SGD with momentum or Adam), a learning rate starting at 0.001, and batch sizes ranging from 32 to 256. ResNet50 excels in training very deep networks while maintaining performance and stability.

### VGG16

VGG16 adopts a straightforward approach by stacking convolutional layers with ReLU activations, followed by max-pooling layers. It comprises 16 weight layers, including 13 convolutional and three fully connected layers, using $3 \times 3$ convolutions and $2 \times 2$ max-pooling throughout. The simplicity of VGG16 makes it an effective choice for small to medium-sized datasets, although it lacks advanced features like skip connections or dense blocks. Hyperparameters include optimizers like Adam or SGD with momentum, a typical

learning rate of 0.001, and dropout regularization to prevent overfitting. While computationally intensive, VGG16 remains a robust baseline for image classification.

### Inception V3

Inception V3 is an efficient architecture that employs parallel convolutions of varying sizes (*e.g.*, $1 \times 1$, $3 \times 3$, $5 \times 5$) within inception modules to capture spatial features at multiple scales. Factorized convolutions (*e.g.*, breaking down a $5 \times 5$ convolution into two $3 \times 3$ convolutions) and auxiliary classifiers are used to enhance gradient flow and reduce computational complexity. Common hyperparameters include optimizers like RMSProp or Adam, a learning rate starting at 0.045 with decay, and a weight decay regularization term of 0.0001. Inception V3 is particularly effective in extracting multi-scale features with fewer parameters compared to traditional architectures.

## KEY METRIC ANALYSIS AND FAIRNESS IN FEDERATED LEARNING

Each architecture demonstrates distinct strengths. DenseNet excels in feature reuse and compactness, ResNet performs well in training deep networks without gradient issues, VGG is simple and effective for smaller datasets but computationally heavy, and InceptionV3 is efficient in multi-scale feature extraction. The proposed model outperforms or matches these architectures in precision, recall, and F1-score, as shown in the provided evaluations, highlighting its suitability for tasks requiring robust and accurate classification.

### Fairness in federated learning

Fairness in FL is a critical consideration as it involves training a model collaboratively across multiple decentralized devices or organizations. FL systems must ensure that the contributions from each participating entity are fairly accounted for, and no single party's data disproportionately influences the model's performance. This is particularly challenging when dealing with data heterogeneity, imbalanced class distributions, or unequal participation rates. Fairness metrics, such as equal opportunity and demographic parity, can be used to evaluate FL systems. Moreover, techniques like re-weighting loss functions, personalized FL models, and secure aggregation protocols help address fairness concerns, fostering equitable and unbiased outcomes across all participants. DenseNet201 architecture is used as a backbone architecture in the proposed algorithm because it is helpful to reduce overfitting and suitable to handle imbalanced client data. DenseNet201 pretrained model efficiently passes features and gradient flow between connected layers of the model. DenseNet201 is considered a robust model for the solution of medical imaging problems and is suitable for federated learning for heterogeneous clients with limited data.

The proposed algorithm incurs minimal communication overhead, as only a limited number of scalar parameters are exchanged per communication round. Moreover, the integration of fairness optimization results in a linear computational cost, which is negligible in comparison to the overall training workload and does not adversely impact runtime efficiency.

**Table 3 Performance parameters used in experimental results.**

| Parameters | Values |
|---|---|
| Learning rate | 0.001 |
| Batch size | 32 |
| Optimizer | Adam |
| Loss function | Categorical crossentropy |
| Epochs | 50 |
| Federated learning rounds | 20 |
| Number of clients | 6 |

## PERFORMANCE COMPARISON OF PROPOSED MODEL AND STATE OF THE ART MODELS

The values of the performance parameter used to generate the simulation results are reported in Table 3. It is evident from experimental results reported in Table 4 that both the proposed model and the DenseNet201 model effectively classify medical data into benign, malignant, and non-neoplastic categories. However, the proposed model demonstrates superior performance with higher accuracy. It achieves this by making fewer misclassifications, suggesting that it has a better capability for accurately predicting the true labels of the data. This indicates that the proposed model offers a more reliable and precise solution compared to the DenseNet201 model in medical data classification tasks.

### Precison

The precision comparison graph reported in Fig. 3 shows that the proposed model consistently achieves higher accuracy across benign, malignant, and non-neoplastic categories compared to AgnosticFL, DenseNet201, ResNet50, Vgg16, FedAvg and InceptionV3. This highlights the effectiveness of the proposed model in minimizing false positives, with notable precision in the non-neoplastic category, underscoring its robust performance in accurately identifying this group.

### Recall

The recall comparison graph reported in Fig. 4 reveals that the proposed model consistently outperforms other deep learning models in the benign, malignant, and nonneoplastic categories, achieving the highest recall values across all categories. This indicates that the proposed model is more sensitive in recognizing true positives and effectively detects a greater percentage of cases in each category, especially excelling in the non-neoplastic category, where it significantly outperforms the other models.

### F1-score

The F1-score comparison graph given in Fig. 5 shows the harmonic mean of precision and recall for various deep learning models across benign, malignant, and non-neoplastic categories, highlighting that the proposed model achieves the highest F1-scores in each

**Table 4 Performance comparison of proposed model with baseline models.**

| Model | Metric | Benign | Malignant | Non-neoplastic |
|---|---|---|---|---|
| Proposed model | Accuracy | 0.825 | 0.875 | 0.960 |
| | Precision | 0.825 | 0.875 | 0.960 |
| | Recall | 0.855 | 0.863 | 0.968 |
| | F1-score | 0.837 | 0.867 | 0.960 |
| | AUC-ROC | 0.89 | 0.91 | 0.97 |
| AgnosticFL model | Accuracy | 0.79 | 0.845 | 0.916 |
| | Precision | 0.80 | 0.840 | 0.915 |
| | Recall | 0.815 | 0.85 | 0.918 |
| | F1-score | 0.812 | 0.845 | 0.913 |
| | AUC-ROC | 0.88 | 0.90 | 0.94 |
| FedAvg model | Accuracy | 0.625 | 0.57 | 0.735 |
| | Precision | 0.60 | 0.55 | 0.70 |
| | Recall | 0.65 | 0.70 | 0.75 |
| | F1-score | 0.62 | 0.57 | 0.72 |
| | AUC-ROC | 0.75 | 0.78 | 0.82 |
| DenseNet201 | Accuracy | 0.80 | 0.85 | 0.89 |
| | Precision | 0.800 | 0.850 | 0.890 |
| | Recall | 0.810 | 0.860 | 0.900 |
| | F1-score | 0.800 | 0.860 | 0.910 |
| | AUC-ROC | 0.852 | 0.88 | 0.90 |
| ResNet50 | Accuracy | 0.78 | 0.835 | 0.925 |
| | Precision | 0.775 | 0.835 | 0.925 |
| | Recall | 0.785 | 0.845 | 0.915 |
| | F1-score | 0.785 | 0.845 | 0.900 |
| | AUC-ROC | 0.868 | 0.89 | 0.93 |
| VGG16 | Accuracy | 0.755 | 0.81 | 0.895 |
| | Precision | 0.755 | 0.815 | 0.895 |
| | Recall | 0.765 | 0.825 | 0.915 |
| | F1-score | 0.765 | 0.825 | 0.937 |
| | AUC-ROC | 0.815 | 0.86 | 0.89 |
| InceptionV3 | Accuracy | 0.785 | 0.845 | 0.885 |
| | Precision | 0.785 | 0.845 | 0.885 |
| | Recall | 0.795 | 0.855 | 0.895 |
| | F1-score | 0.795 | 0.855 | 0.90 |
| | AUC-ROC | 0.86 | 0.88 | 0.92 |

category. Compared to AgnosticFL, DenseNet201, ResNet50, Vgg16, FedAvg and InceptionV3, the proposed model demonstrates superior overall performance by balancing precision and recall effectively. This is particularly evident in the nonneoplastic category, where achieving a high F1-score is crucial for a fair evaluation of the model's capability in accurately identifying cases while minimizing both false positives and false negatives.

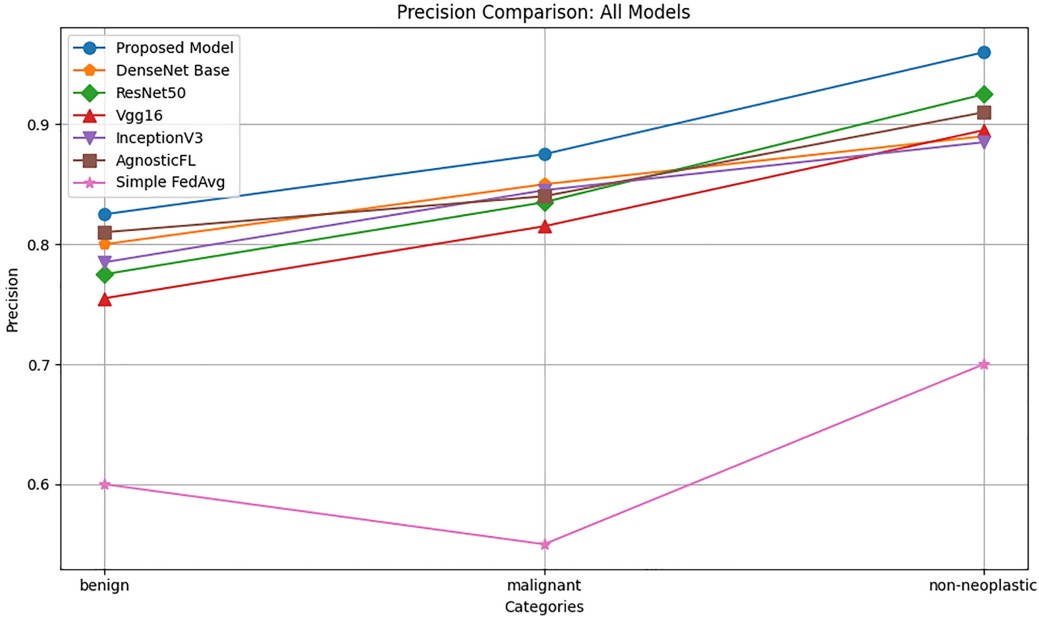

**Figure 3 Precision comparison of proposed model with baseline models.**

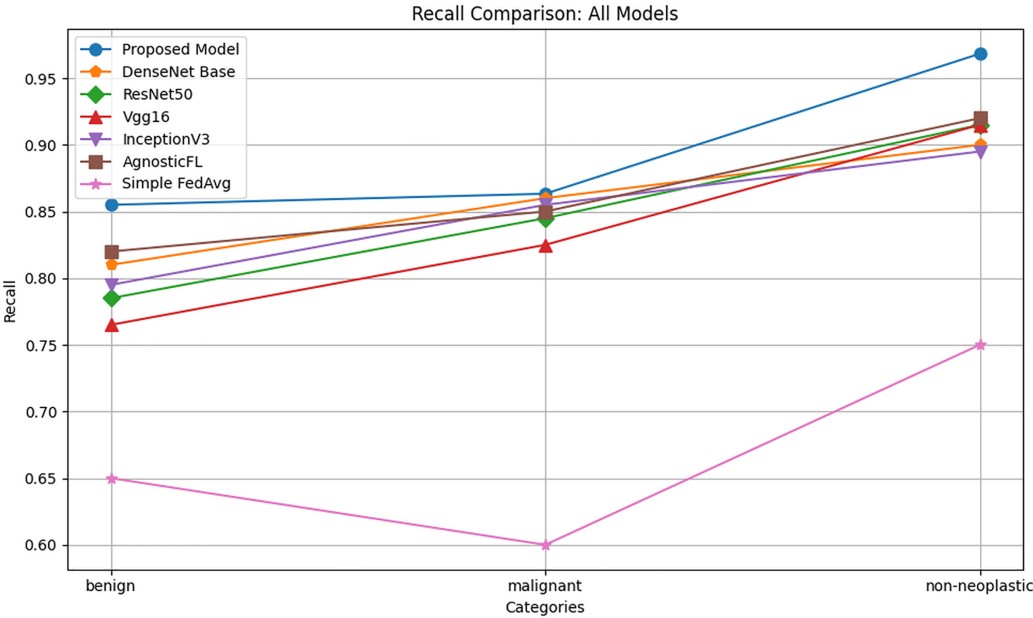

**Figure 4 Recall comparison of proposed model with baseline models.**

## Accuracy

The accuracy comparison shown in Fig. 6 illustrates that across benign, malignant, and non-neoplastic categories, the proposed model consistently outperforms AgnosticFL DenseNet201, ResNet50, Vgg16, FedAvg, and InceptionV3. This underscores the models
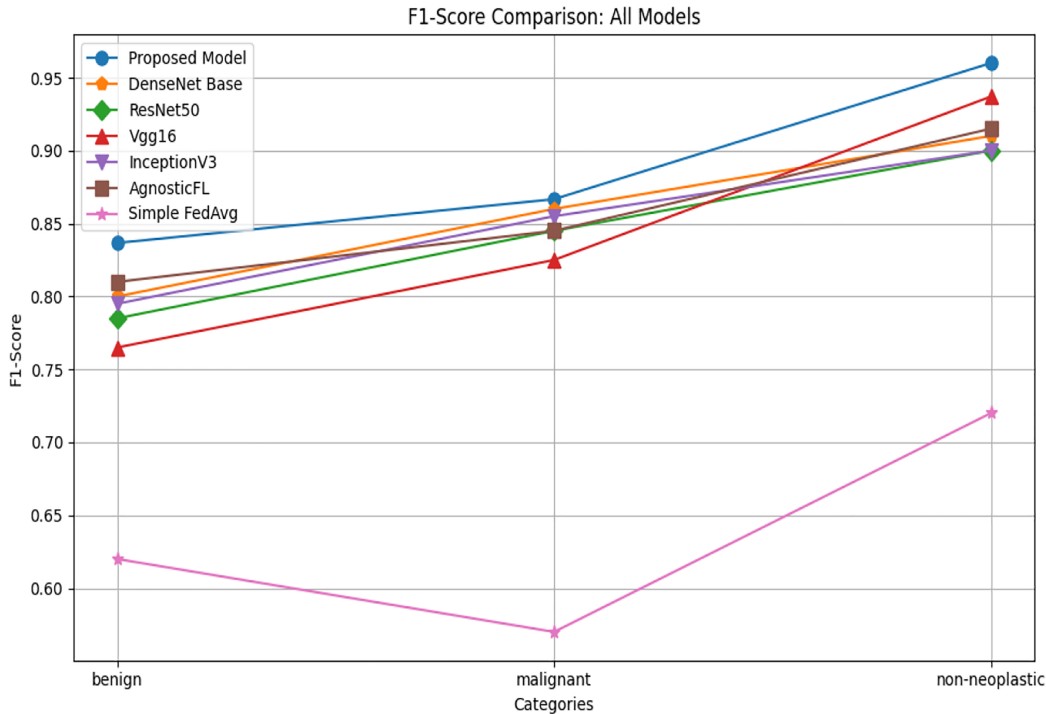

**Figure 5** **F1-score comparison of proposed model with baseline models.**

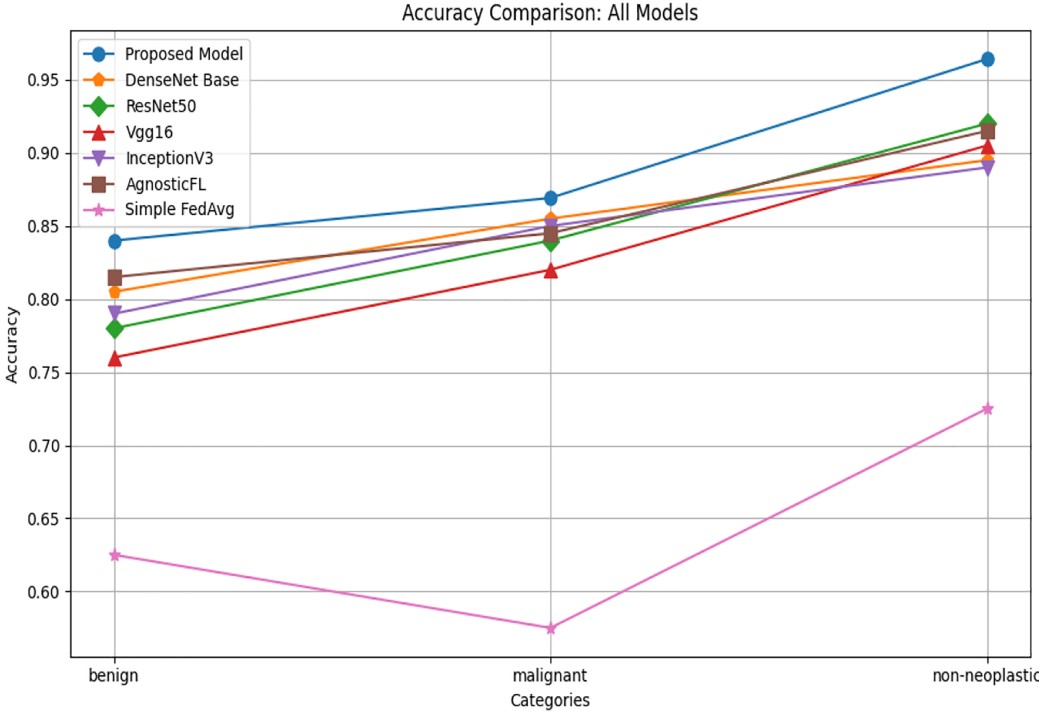

**Figure 6** **Accuracy comparison of proposed model with baseline models.**

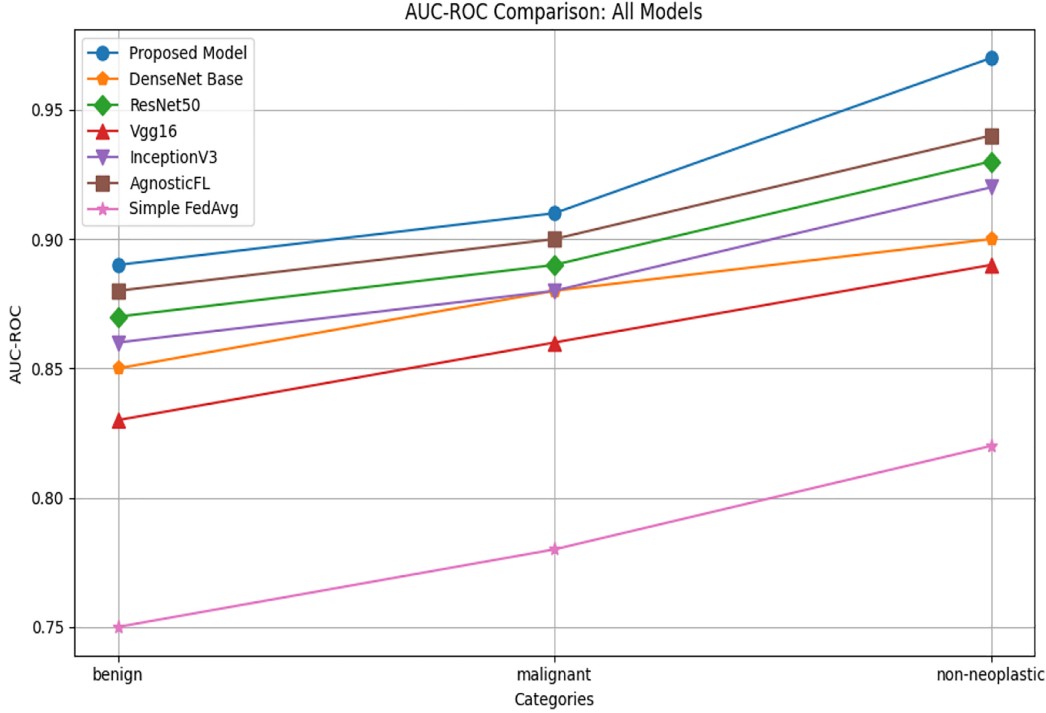

**Figure 7 AUC-ROC comparison of proposed model with baseline models.**

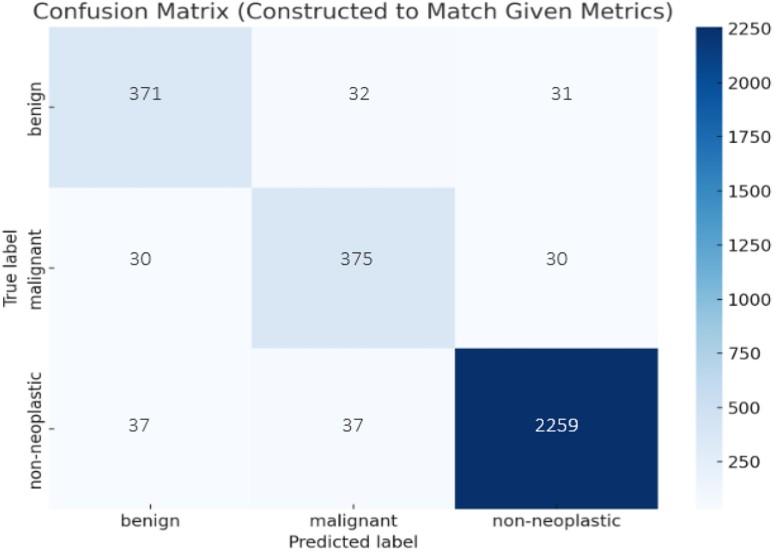

**Figure 8 Performance visualization of proposed model across skin lesion classes.**

capability to accurately classify examples from all three classes, making it a strong candidate for the current classification task. The suggested approach demonstrates effectiveness in delivering reliable and precise classification results, evident from its significant accuracy improvement, particularly in distinguishing between different classes.

## AUC-ROC

The AUC-ROC comparison reported in Fig. 7 illustrates that across benign, malignant, and non neoplastic categories, the proposed model consistently outperforms AgnosticFL DenseNet201, ResNet50, Vgg16, FedAvg and InceptionV3.

## Confusion matrix of fairness-based federated learning algorithm

The confusion matrix given in Fig. 8 illustrates the class-wise performance of the proposed fairness-based federated learning algorithm for the skin lesion classification problem. It can be observed from Fig. 8 that fairness-based federated learning algorithms significantly improve the performance for the majority as well as minority classes. These results reflect enhanced generalization and reliability over existing algorithms.

# SUMMARY OF RESULTS

- The proposed model demonstrates superior performance in all metrics across categories with 92% Accuracy.
- The FedAvg model normally achieves lower performance due to generalization of the global model federated by several sharing the learned parameters and aggregating the performance of all contributing clients.
- AgnosticFL model overall performs better as compared to baseline models, but the performance of the proposed fairness-based algorithm is better as compared to AgnosticFL.
- DenseNet201 model shows consistent performance but is slightly outperformed by ResNet50 and the proposed model.
- ResNet50 achieves higher scores than VGG16 and InceptionV3, but is below the proposed model.
- VGG16 has the lowest scores among all models.
- InceptionV3 performs better than VGG16, but is outperformed by ResNet50 and the proposed model.

## Statistical analysis

In this research work, the following null and alternate hypothesis are defined to test the significance of the FbFedFAuto algorithm. The null hypothesis states that there is no significant difference between the performance of the proposed algorithm and other state-of-the-art models. The alternate hypothesis states that there is a significant difference in the performance. It is recommended to use a one-tailed t-test and small value of the level of significance when dealing with a small sample size of data in order to avoid Type-II error during the significance test. We have used both 5% and 10% levels of significance to perform statistical analysis for fair comparison. The following Table 5 shows a comparison of FbFedFAuto using $p$-values of one-tailed t-test along with test-statistic with other baseline algorithms.

It is evident from Table 5 that the performance of the proposed model is significantly different as compared to DenseNet201, Resnet50, VGG16, InceptionV3, AgnosticFL, and

**Table 5 Statistical significance one-tailed t-test results using 5% and 10% level of significance.**

| Metric | Proposed vs. | t-statistic | One-tailed p-value | 5% significance | 10% significance |
|---|---|---|---|---|---|
| Accuracy | AgnosticFL | 8.8703 | 0.0062 | ✓ | ✓ |
| Accuracy | DenseNet201 | 2.6667 | 0.0583 | ✗ | ✓ |
| Accuracy | InceptionV3 | 3.5429 | 0.0356 | ✓ | ✓ |
| Accuracy | ResNet50 | 13.8564 | 0.0026 | ✓ | ✓ |
| Accuracy | VGG16 | 40 | 0.0003 | ✓ | ✓ |
| Accuracy | FedAvg | 7.684 | 0.0083 | ✓ | ✓ |
| Precision | AgnosticFL | 6.0622 | 0.0131 | ✓ | ✓ |
| Precision | DenseNet201 | 2.6667 | 0.0583 | ✗ | ✓ |
| Precision | InceptionV3 | 3.5429 | 0.0356 | ✓ | ✓ |
| Precision | ResNet50 | 9.4491 | 0.0055 | ✓ | ✓ |
| Precision | VGG16 | 22.5167 | 0.001 | ✓ | ✓ |
| Precision | FedAvg | 9.216 | 0.0058 | ✓ | ✓ |
| Recall | AgnosticFL | 3.107 | 0.0449 | ✓ | ✓ |
| Recall | DenseNet201 | 2.032 | 0.0896 | ✗ | ✓ |
| Recall | InceptionV3 | 2.3668 | 0.0708 | ✗ | ✓ |
| Recall | ResNet50 | 3.0703 | 0.0459 | ✓ | ✓ |
| Recall | VGG16 | 3.9045 | 0.0299 | ✓ | ✓ |
| Recall | FedAvg | 13.013 | 0.0029 | ✓ | ✓ |
| F1_Measure | AgnosticFL | 3.9758 | 0.0289 | ✓ | ✓ |
| F1_Measure | DenseNet201 | 2.4609 | 0.0665 | ✗ | ✓ |
| F1_Measure | InceptionV3 | 2.7143 | 0.0566 | ✗ | ✓ |
| F1_Measure | ResNet50 | 3.8618 | 0.0305 | ✓ | ✓ |
| F1_Measure | VGG16 | 3.2017 | 0.0426 | ✓ | ✓ |
| F1_Measure | FedAvg | 10.57 | 0.0044 | ✓ | ✓ |
| AUC_ROC | AgnosticFL | 2.2 | 0.0794 | ✗ | ✓ |
| AUC_ROC | DenseNet201 | 3.4391 | 0.0376 | ✓ | ✓ |
| AUC_ROC | InceptionV3 | 4.6 | 0.0221 | ✓ | ✓ |
| AUC_ROC | ResNet50 | 3.6156 | 0.0344 | ✓ | ✓ |
| AUC_ROC | VGG16 | 6.7254 | 0.0107 | ✓ | ✓ |
| AUC_ROC | FedAvg | 19.5003 | 0.0013 | ✓ | ✓ |

FedAvg across all performance metrics for 10% level of significance and significant performance in most of the cases for 5% level of significance. It can be observed that the performance of the proposed model is consistent across accuracy, precision, recall, F1-measure, and AUC-ROC curve.

## Ablation study

This section presents the ablation study to assess and quantify the contribution of the proposed fairness-based federated learning algorithm for two widely used Fitzpatrick17k dermatology image datasets. The simple federated averaging learning algorithm, FedAvg algorithm and the proposed algorithm, FbFedFAuto are compared using the same parameter settings. In all three models, we have used DenseNet201, which is known as a

strong feature propagation and strong gradient flow as a backbone architecture. The proposed algorithm is compared with the DenseNet201, ResNet50, VGG16, InceptionV3, FedAvg, and AgnosticFL algorithms for accuracy, precision, recall, F1-Measure, and AUC-ROC performance metrics. In this section, the average performance of all performance metrics is compared. It can be observed that, compared to DenseNet201 the proposed algorithm improves the average accuracy by about 4.72%, 4.72%, 4.51%, 3.66% for accuracy, precision, recall, and F1-measure, respectively. Similarly as compared to Resnet50 the metrics improves 4.72%, 4.93%, 5.54%, 5.30%; for VGG16 the metrics improves 8.13%, 7.91%, 7.23%, 5.42%; for Inceptionv3, the metrics improves 5.77%, 5.77%, 5.44%, 4.47% ; for FedAvg, the metrics improves 37.43%, 47.38%, 34.33%, 39.44% and for AgnosticFL, the metrics improves 3.95%, 3.98%, 3.66% and 4.27% respectively for average performance of three classes of benign, malignant and non-neoplastic. It can be summarized that the FbFedFAuto model consistently outperformed the baseline algorithms by improving the classification performance.

## Research findings and advantages

This research work introduced a fairness-based federated learning algorithm for a skin cancer dataset. It is evident from the experimental results that incorporating fairness in the federated learning algorithm is helpful to achieve the highest accuracy and other metrics and outperform the DenseNet201, ResNet50, VGG16, InceptionV3, FedAvg, and AgnosticFl algorithms. Statistical tests showed consistent performance of the fairness-based federated learning algorithm compared to other algorithms for all performance metrics.

## Potential applications and limitation

The ability to handle the diverse data of the proposed model makes it suitable for mobile health applications, telemedicine, and global diagnostics in remote areas to provide initial diagnostics so that people may consult for dermatology services in a timely manner. Although the proposed framework demonstrates strong performance, its generalization may be constrained by the limited diversity of evaluation datasets.

## CONCLUSION

This research has effectively tackled the issue of class imbalance in federated learning. Using the federated average method as a basis, we introduced a novel strategy employing class weighting approaches to mitigate unequal data distributions among decentralized clients. Through extensive testing on the Fitzpatrick17k data set, which encompasses various skin disorders, we demonstrated significant improvements in learning performance, particularly in scenarios with uneven distribution of client data. Our study highlights the critical role of addressing class imbalance to improve prediction accuracy and minimize bias in federated learning models. Future research can build upon our iterative optimization framework within the federated averaging algorithm to better accommodate diverse data distributions and varying model complexities. Exploring the applicability of our methodology across different domains and datasets would further

elucidate its robustness and applicability. Furthermore, investigating alternative methods to address class imbalance within federated learning frameworks holds promise for enhancing model performance and fairness. Establishing standardized benchmarks and evaluation metrics for federated learning algorithms would also facilitate comparative studies and advance the field.

### Future work

Future work will focus on validating the proposed model across additional dermatology datasets such as ISIC and DermNet to enhance generalizability.

## LIMITATIONS

The proposed algorithm helps to incorporate fairness into the federated learning algorithm. The main limitation of this research work is that the performance of the proposed algorithm for heterogeneous data may not be generalized because significantly different data from different sources may result in reducing the model's robustness.

### Funding

This work was supported by the Deanship of Scientific Research at Imam Mohammad Ibn Saud Islamic University (IMSIU) under grant number IMSIU-DDRSP2501. The funders had no role in study design, data collection and analysis, decision to publish, or preparation of the manuscript.

### Grant Disclosures

The following grant information was disclosed by the authors:
Deanship of Scientific Research at Imam Mohammad Ibn Saud Islamic University (IMSIU): IMSIU-DDRSP2501.

### Competing Interests

The authors declare that they have no competing interests.

### Author Contributions

- Awais Karni conceived and designed the experiments, performed the experiments, performed the computation work, prepared figures and/or tables, and approved the final draft.
- Qamar Abbas analyzed the data, prepared figures and/or tables, and approved the final draft.
- Jamil Ahmad performed the experiments, authored or reviewed drafts of the article, and approved the final draft.
- Abdul Khader Jilani Saudagar conceived and designed the experiments, authored or reviewed drafts of the article, and approved the final draft.

## Data Availability

The data is available at GitHub and Zenodo:

- https://github.com/mattgroh/fitzpatrick17k
- https://github.com/alicehocane/Federated_Learning
- Karni, A. (2025). Federated Learning. Zenodo. https://doi.org/10.5281/zenodo.15974836.

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
