# Peer review of "Skin cancer classification using novel fairness based federated learning algorithm"

_PeerJ Computer Science, doi:10.7717/peerj-cs.3171_

## Round 0.1 · original submission · Major Revisions

Dear authors,

You are advised to critically respond to all comments point by point when preparing an updated version of the manuscript and while preparing for the rebuttal letter. Please address all comments/suggestions provided by reviewers, considering that these should be added to the new version of the manuscript.

Kind regards,
PCoelho

Reviewer 1 ·

Basic reporting

The proposed "Fairness-Based Automatic Weight Adjusting FL (FbFedFAuto)" is not sufficiently novel. It is essentially a variation of FedAvg with fairness-aware reweighting using standard deviation-based adjustments.

There is no comparative evaluation against other fairness-aware federated learning algorithms like FedProx, FedNova, FedBN, or AgnosticFL, despite citing them.

The paper includes repetitive phrases and awkward constructions (e.g., “we have equally divided our into 20 rounds”), which hurt readability.

Sentences are often verbose and could be rewritten more concisely for scientific clarity.

The abstract and conclusion overstate the impact: a 92% accuracy on a single dataset without generalization testing does not support claims of clinical readiness.

The phrase “a scalable and accurate solution for real-world applications” is premature and unsupported.

Experimental design

Novelty is limited.

Only basic metrics (Precision, Recall, F1-score, Accuracy) are reported.

No statistical significance (e.g., p-values, confidence intervals, standard deviation) is provided, and no ROC/AUC curves are shown.

The contribution of the fairness weighting mechanism is not isolated through ablation.

There is no baseline performance of standard FedAvg, nor a comparison with FbFedFAuto without the fairness adjustment.

Validity of the findings

Authors use both Google Colab and a local HP laptop, but no clarity is given on which experiments ran where.

The use of DenseNet201 across clients is claimed, but hardware specifications suggest limitations, casting doubt on how realistic it is to train such a deep model across clients on limited devices.

The model is only evaluated on the Fitzpatrick17k dataset, which limits generalizability. This dataset, while diverse in skin tones, does not reflect the real-world diagnostic environment (e.g., hospital-acquired dermoscopy images).

There is no external dataset (such as ISIC or DermNet) used for independent validation.

Reviewer 2 ·

Basic reporting

Minimal Introduction: The introduction section lacks depth and fails to establish a strong context or motivation for the research. Please elaborate on the problem statement, domain relevance, and provide a clear narrative that leads to your research objectives.

Placement of Significance Statement: The significance of your work is currently placed in the introduction, which is premature and lacks evidence. It would be more appropriate and convincing to present this after the Results and Discussion section, with supporting data and interpretations.

Incomplete Materials Section: The Materials section omits crucial information about the dataset(s) used. Please include detailed dataset characteristics such as size, class distribution, preprocessing steps, and source.

Misplaced Reproducibility Statement: The reproducibility statement is placed in the middle of the manuscript, which interrupts the logical progression. Kindly relocate this section just before the references, as per standard practice.

Experimental design

Vague Research Goals: The research goals and objectives are overly generic. There is no clarity on the specific techniques, methodologies, or unique contributions. Clearly articulate what sets your work apart and highlight the novelty of your approach.

Disorganized Literature Review: The entire literature review is presented as one long paragraph, making it difficult to comprehend. Organize the review into thematic or methodology-based subsections to improve readability and logical flow.

Underdetailed Methods Section: The Methods section lacks a structured, step-by-step breakdown of the proposed approach. Avoid vague descriptions and provide a clear, phase-wise or modular explanation of your methodology.
Missing Dataset Citation and Visualization: The Fitzpatrick 17k dataset is not properly cited, and no visual samples are shown. Include the proper citation and sample images to support transparency and aid understanding of the dataset.

Incoherent Framework Illustration: Figure 1 does not align with the goals outlined in the introduction. While privacy-preserving and robustness are claimed, there is no demonstration of how these are addressed. Additionally, the architecture appears generic with no evident novelty, leading to contradictory claims.

Validity of the findings

Unsupported Algorithm Claim: The statement that your federated learning algorithm ensures fairness and maintains performance is unsubstantiated. You must justify this claim through ablation studies, fairness metrics, and empirical comparisons.

Repetition in Experimental Setup: The Experimental Setup section largely repeats content already mentioned in Materials, without added value. Consolidate this content and organize it logically to avoid redundancy.

Missing Hyperparameter Table: Hyperparameters and a model summary are essential for reproducibility and transparency. Present these details explicitly in a well-organized table.
Mismatch Between Objectives and Results: The research objectives focus on handling imbalanced datasets, yet no such experiments or evaluation methods are evident in the results (Figures 2-5, Table 2). Explain this discrepancy clearly.

Redundant and Unsupported Objectives: The objectives related to imbalanced datasets are repetitive and unsubstantiated. There is no explanation or illustration of class imbalance, nor any strategy (before/after) for handling it. Clarify this with proper experimentation and statistical insights.

Unjustified Accuracy Claim: The claim of achieving 92% accuracy is unconvincing without supporting evidence such as a confusion matrix, accuracy/loss plots, and comparative metrics. It is also unclear whether your baseline comparisons are fair or ethical, particularly given the lack of transparency in the experimental setup.

Additional comments

Very weak and synthesised paper - where the objective architecture and results have no connections. Not to the standard of a research paper

Reviewer 3 ·

Basic reporting

-

Experimental design

-

Validity of the findings

-

Additional comments

This paper is about skin cancer classification, and my suggestions about this paper are:
Clarity of Writing: The manuscript is generally understandable, but contains occasional typographical and grammatical errors (e.g., “classiûcation” in the title, inconsistent hyphenation) that should be corrected for a professional presentation.

Introduction & Motivation: The Introduction provides a solid overview of federated learning and class‐imbalance challenges, but could better highlight how the proposed fairness weighting differs from or improves upon existing fairness‐aware FL methods in the literature.
Literature Coverage: References are comprehensive, yet more discussion is needed to position this work relative to closely related fairness‐focused FL algorithms (e.g., AgnosticFair, Auto‐FedAvg).

You can analyze the papers below.
- A lightweight deep convolutional neural network model for skin cancer image classification
- New pyramidal hybrid textural and deep features-based automatic skin cancer classification model: Ensemble DarkNet and textural feature extractor
- Skin cancer classification model based on hybrid deep feature generation and iterative mRMR

Experimental Design
Dataset & Preprocessing: Use of the Fitzpatrick17k dataset is appropriate; however, details on how images were partitioned among simulated clients (e.g., non‐IID splits, number of clients) are insufficiently described.

Method Description: Algorithm 1 clearly outlines the fairness weighting mechanism, but variable definitions (e.g., λc, m, M) should be summarized in a table or the main text for easier reference.
Baseline Comparisons: Comparing against DenseNet, ResNet50, VGG16, and InceptionV3 is comprehensive; yet, the choice of hyperparameter settings for each baseline should be specified to ensure reproducibility

Discussion:
Authors should discuss
- Findings
- Advantages
- Limitations and future works
- Potential applications

Reviewer 4 ·

Basic reporting

-

Experimental design

-

Validity of the findings

-

Additional comments

1. How does the proposed method scale with the number of clients, particularly when clients have drastically different data volumes or entirely missing classes?
2. Why was only DenseNet201 used as the core architecture for the federated learning training, despite a comparison to other models?
3. The paper does not detail if the randomness sources (e.g., client data shuffling, initialization seeds, augmentation parameters) were controlled during experiments.
4. To enrich the manuscript, more related works could be reviewed, such as Skin cancer classification using fine-tuned transfer learning of DENSENET-121, 10.3390/app14177707

5. The use of DenseNet201 is sound due to its compactness and feature reuse, but the architectural bias might mask the actual benefit of the fairness algorithm.
6. What are the computational and communication costs of the proposed algorithm relative to FedAvg and other fairness-aware FL schemes?
7. To what extent does the proposed approach generalize beyond the Fitzpatrick17k dataset?

---

## Round 0.2 · accepted · Accept

Dear authors, we are pleased to verify that you meet the reviewer's valuable feedback to improve your research.

Thank you for considering PeerJ Computer Science and submitting your work. The citations requested by Reviewer 2 are not needed.

Kind regards
PCoelho

Reviewer 2 ·

Basic reporting

Cite the recent relevant research suggested below to align with the objective, which strengthens the research

Veeramani, N., & Jayaraman, P. (2024). YOLOv7‐XAI: Multi‐Class Skin Lesion Diagnosis Using Explainable AI With Fair Decision Making. International Journal of Imaging Systems and Technology, 34(6), e23214.
Veeramani, N., Jayaraman, P., Krishankumar, R., Ravichandran, K. S., & Gandomi, A. H. (2024). DDCNN-F: double decker convolutional neural network'F'feature fusion as a medical image classification framework. Scientific Reports, 14(1), 676.
Nirmala, V., Shashank, H. S., Manoj, M. M., Satish, R. G., & Premaladha, J. (2023). Skin Cancer Classification Using Image Processing with Machine Learning Techniques. In Intelligent Data Analytics, IoT, and Blockchain (pp. 1-15). Auerbach Publications.
Jayaraman, P., Veeramani, N., Krishankumar, R., Ravichandran, K. S., Cavallaro, F., Rani, P., & Mardani, A. (2022). Wavelet-based classification of enhanced melanoma skin lesions through deep neural architectures. Information, 13(12), 583.

Experimental design

No comments. All are addressed thoroughly by the authors

Validity of the findings

Appropriate

Reviewer 4 ·

Basic reporting

no comment

Experimental design

no comment

Validity of the findings

no comment

Additional comments

My concerns have been well-addressed.